# Membership Attacks on Conditional Generative Models Using Image Difficulty

## Abstract

Membership inference attacks (MIA) try to detect if data samples were used to train a neural network model. As training data is very valuable in machine learning, MIA can be used to detect the use of unauthorized data. Unlike the traditional MIA approaches, addressing classification models, we address conditional image generation models (e.g. image translation). Due to overfitting, reconstruction errors are typically lower for images used in training. A simple but effective approach for membership attacks can therefore use the reconstruction error. However, we observe that some images are "universally" easy, and others are difficult. Reconstruction error alone is less effective at discriminating between difficult images used in training and easy images that were never seen before. To overcome this, we propose to use a novel difficulty score that can be computed for each image, and its computation does not require a training set. Our membership error, obtained by subtracting the difficulty score from the reconstruction error, is shown to achieve high MIA accuracy on an extensive number of benchmarks.

## 1 Introduction

Deep neural networks have been widely adopted in various computer vision tasks, e.g. image classification, semantic segmentation, image translation and generation etc. The high sample-complexity of such models requires large amounts of training data. However, obtaining many training images might not be an easy task. In fact, collection and annotation is often an expensive and labor intensive process. In some domains, such as medical imaging, publicly available training data are particularly scarce due to privacy concerns. In such settings, it is common to grant access to private sensitive data for training purposes alone, while ensuring not to reveal the data in the inference stage. A common solution is training the model privately and then providing black-box access to the trained model. However, even black-box access may leak sensitive information about the training data.

*Membership inference attacks (MIA)* are one way to detect such leakage. Given access to a data sample, an attacker attempts to find whether or not the sample was used in the training process. MIA attacks have been widely studied for image classification models, achieving high success rates (Shokri et al., 2017; Salem et al., 2018; Sablayrolles et al., 2019; Yeom et al., 2018; Li & Zhang, 2020; Choo et al., 2020). Due to overfitting in deep neural networks, prediction confidence tends to be higher for images used in training. This difference in prediction confidence helps MIA methods to successfully determine which images were used for training. Therefore, in addition to detecting information leakage, MIA also provide insights on the degree of overfitting in the victim model.

We address MIA for a new domain - conditional image generation models, e.g. image translation. While classification models give a probability vector over possible classes, generation models give a single color for every pixel. We propose a MIA that uses pixel-wise reconstruction error, as overfitting causes lower reconstruction error on images used for training. But we observe that some images are "universally" easy, and others are universally difficult. Reconstruction error alone is therefore less accurate at discriminating between difficult images used in training and previously unseen easy images. To overcome this limitation, we add a novel image difficulty score which is computed for each query image. Our image difficulty score uses the accuracy of a linear predictor computed over a given image, predicting pixel values from deep features of that image. The reconstruction error together with the difficulty score helps to discriminate between two factors of variation in the reconstruction error, namely (i) The "intrinsic" difficulty of the conditional generation task for each image,

based on its difficulty score and (ii) The boost in accuracy due to overfitting to the training images. Defining a membership error that subtracts the difficulty score from the reconstruction error is shown empirically to achieve high success rates in MIA. Differently from other MIA approaches, we do not assume the existence of a large number of in-distribution data samples for training a shadow model - but rather operate on merely a single image. Our method is evaluated on an extensive number of benchmarks demonstrating its effectiveness compared to strong baseline methods.

## 2 RELATED WORK

### 2.1 MEMBERSHIP INFERENCE ATTACKS (MIA)

Shokri et al. (2017) were the first to study MIA against classification models in a black-box setting. In black-box setting the attacker can only send queries to the victim model and get the full probability vector response, without being exposed to the model itself. They proposed to train multiple shadow models to mimic the behavior of the victim model, and then use those to train a binary classifier to distinguish between known samples from the train set and unknown samples. They assume the existence of in-distribution new training data and knowledge of the victim model architecture.

Salem et al. (2018) further relaxed those assumptions and demonstrated that using only one shadow model is sufficient for a successful attack, and proposed using out-of-distribution dataset and different shadow model architectures, for a slightly inferior attack. Even more interestingly, they showed that without any training, a simple threshold on the victim model's confidence score is sufficient. This shows that classification models are more confident of samples that appeared in the training process, compared to unseen samples.

Sablayrolles et al. (2019) proposed an attack based on applying a threshold over the loss value rather then the confidence and showed that black-box attacks are as good as white-box attacks. As the naive defense against such attacks is to modify the victim model's API to only output the predicted label, other works proposed label-only attacks (Yeom et al., 2018; Li & Zhang, 2020; Choo et al., 2020).

While most previous work has been around classification models, there has been some effort regarding MIA on generative models such as GANs and VAEs (Chen et al., 2019; Hayes et al., 2019; Hilprecht et al., 2019). An attack against semantic segmentation models was proposed by He et al. (2019), where a shadow semantic segmentation model is trained, and is used to train a binary classifier. The classifier is trained on image patches, and the final decision regarding the query image is set by the aggregation of the per-patch classification scores. The input to the classifier is a structured loss map between the shadow model's output and the ground truth segmentation map. Although this task is the closest to ours, our work is the first study of membership inference attacks on conditional image generation model.

Besides membership inference attacks, other privacy attacks against neural networks exist. We refer the reader to Sec. A.1 for more details of such attacks.

### 2.2 CONDITIONAL IMAGE GENERATION

Image-to-image translation is the task of mapping an image from a source domain to a target domain, while preserving the semantic and geometric content of the input image. Over the last decade, with the advent of deep neural network models and increasing dataset sizes, significant progress was made in this field. Currently, the most popular methods for training image-to-image translation models use Generative Adversarial Networks (GANs) (Goodfellow et al., 2014) and are currently used in two main scenarios: (i) unsupervised image translation between domains (Zhu et al., 2017a; Kim et al., 2017; Liu et al., 2017; Choi et al., 2018); (ii) serving as a perceptual image loss function (Isola et al., 2017; Wang et al., 2018; Zhu et al., 2017b). In this work we introduce the novel task of MIA on conditional image generation models.

## 3 MIA ON CONDITIONAL IMAGE GENERATION MODELS

In membership inference attacks (MIA), an adversary attacks a victim model by attempting to infer whether a query data sample was used to train a victim model. Such attacks exploit overfitting to

the training data performed by the victim model. For classification, an overfitted model will likely be more confident of the prediction for data samples that were included in the training set.

We focus on MIA in a new domain: conditional image generation models, e.g. image translation. We propose a simple and effective attack. Differently from most previous works (Shokri et al., 2017; He et al., 2019), we do not use shadow models or train a binary classifier, and thus do not require any additional training data. We assume the most restrictive attack setting, where the attacker only has black-box access to the victim model $\mathbf{V}$, and has no knowledge of its weights and architecture.

Our membership attack is performed on a pair of query images $(x, y)$ where $x$ is an image from the input domain and $y$ is the ground truth from the output domain. The existence of the ground truth image $y$ is in-line with previous works, and is a reasonable assumption in conditional generative models. For each query we compute a membership error, $L_{mem}$ (see Eq. (3)), to which we apply a pre-defined threshold $\tau$, such that all queries where $L_{mem}(x, y) < \tau$ are marked as members of the training data. The membership error has two elements: reconstruction error and difficulty score.

## 3.1 RECONSTRUCTION ERROR FOR MEMBERSHIP EVALUATION

Typical MIA that operate on classification models consider the probability (or confidence) given by the model to the correct class. Image generation models are different as they output a color value of each pixel. This value is the maximum likelihood estimate, and no probability distribution over possible values is given. As the confidence of the model prediction is unknown, we propose to examine the reconstruction error of the model over all pixels.

The reconstruction error is described in Eq. (1), and is computed using the $L_1$ error between the output image predicted by a black-box access to the model, $\mathbf{V}(x)$, and the ground truth image $y$. We show that for images in the training set, the model output has mostly lower prediction errors compared to unknown images. See Table 1.

$$L_{rec}(x, y) = \|\mathbf{V}(x) - y\|_1 \tag{1}$$

## 3.2 EASY IMAGES AND DIFFICULT IMAGES

In this section, we tackle the following question: Given an image, compute a score that measures how "difficult" it is to synthesize it. Consider, for example, the task of supervised segmentation-to-image translation. I.e. the task is to "invert" the segmentation process, and recover the original image that gave rise to a given segmentation map. It is clear that not all images are equally difficult: (i) more difficult images have sharp and detailed textures whereas simpler images have blurrier textures; (ii) images with semantic segmentation maps that contain only few categories provide less guidance than those with more detailed segmentation maps, making the correct prediction less certain. Image difficulty score should quantify these difficulties. In Sec. 4.1 we show that such a difficulty score is important to increase accuracy of membership inference attacks on conditional generative models.

We briefly describe two previous approaches for measuring image difficulty:

**Human-Supervised:** Tudor Ionescu et al. (2016) proposed to define image difficulty as the human response time for solving a visual search task. For this, they collected human annotations for the PASCAL VOC 2012 dataset (Everingham et al., 2010) and trained a regression model, based on pre-trained deep features, to predict the collected difficulty score. The disadvantage of this method is that human-specified difficulty scores may not correlate to the difficulty of image synthesis by neural networks. This is demonstrated empirically in Sec. 4.3.

**Multi-Image:** Another approach taken by Chen et al. (2019) is training a generative model on a set of images similar to the target image distribution. Consider for example training an autoencoder on a set of external images. This approach uses the reconstruction error on the target image as its difficulty score - larger reconstruction errors correspond to more difficult images. This approach has a significant drawback: a large number of images, similar to the target image, are required in order to learn a reliable generative model. In many cases, images from the target distribution may not be available. Additionally, training a model for every task is tedious and computationally expensive.

**Proposed - Single-image difficulty score:** We propose a novel method to assign a difficulty score for image generation models. The difficulty score measures the success of a linear regression model

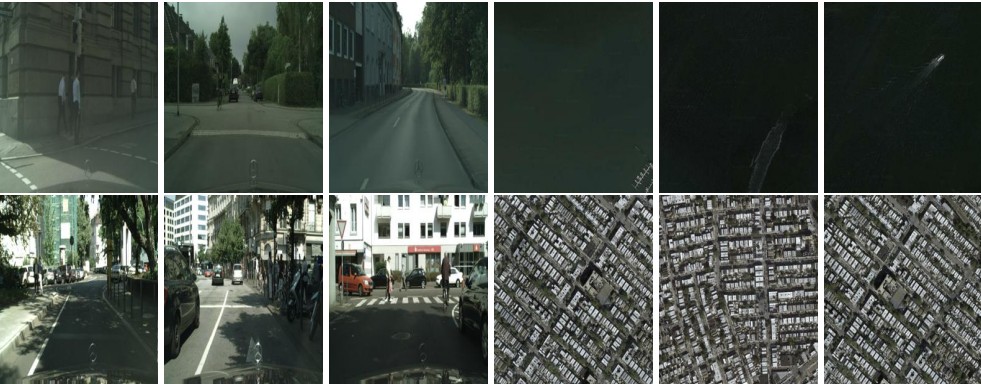

Figure 1: Examples of images from the Cityscapes and Maps2sat datasets that received the lowest (first row) and highest (second row) image difficulty scores using our single-image approach. It can be seen that detailed images with complicated patterns are ranked as difficult, while images with less details and lower contrast are ranked as easier.

to predict the pixel values from a high-level representation of the image. A related approach was proposed by Hacohen & Weinshall (2019) for measuring image difficulty for classification models. Our method is significantly different as it is trained on a single image rather than a large dataset, and that it focuses on generation rather than classification.

Our features are the activation values in the first $4$ blocks of a pre-trained Wide-ResNet50$\times$2 (Zagoruyko & Komodakis, 2016), concatenated together, giving $56\times56$ feature vectors of size $3840$. We reduce the input image to $56\times56$ to match the spatial dimension of the first Wide-ResNet50$\times$2 block. We denote the concatenated feature vector for pixel $i$ as $\psi(i)$. See Sec. A.2 for more details.

The linear regression model $\mathbf{P}$ is a matrix of size $3840\times3$, multiplied with the feature vector $\psi(i)$ of pixel $i$ to give a linear estimate of the RGB colors $y^i$. We minimize $\mathbf{P}$ over 70% randomly selected pixels. The image difficulty score is the average absolute error over the 30% unselected pixels:

$$L_{diff}(x,y) = \frac{1}{N}\sum_{i=1}^{N}\|\mathbf{P}\psi(i) - y^i\|_1 \tag{2}$$

where $y^i$ is the ground truth value of the $i_{th}$ pixel in the resized ground truth image $y$. Fig. 1 presents examples of images that received the highest and lowest difficulty scores.

### 3.3 MEMBERSHIP ERROR

As observed before, some images are "universally" easy to reconstruct, while others are universally difficult to reconstruct. While the reconstruction error in Eq. 1 achieves high MIA success rates, it has a significant limitation - it does not discriminate between difficult and easy samples. Difficult training samples might be more difficult for the victim model to generate and therefore can receive high reconstruction error. Similarly, an easy unknown sample can be generated by the victim model with lower error. Such cases can cause wrong classification if only the reconstruction error is used.

Given our image difficulty score $L_{diff}$ in Eq. (2) and the reconstruction error $L_{rec}$ Eq. (1) we calculate a membership error $L_{mem}$ as follows:

$$L_{mem}(x,y) = L_{rec}(x,y) - \alpha \cdot L_{diff}(x,y) \tag{3}$$

$L_{mem}$ is computed by subtracting the difficulty score $L_{diff}$ from the reconstruction error $L_{rec}$ weighted by $\alpha$ (unless specified otherwise, we use $\alpha = 0.5$). This lowers the membership error $L_{mem}$ for harder-to-predict images compared to easier-to-predict images having the same reconstruction error. See Fig. 2 for an overview illustration of our method.

Using the membership error $L_{mem}$ for MIA substantially improves the success rates in all of our experiments, as shown in Table 1 and Fig. 3

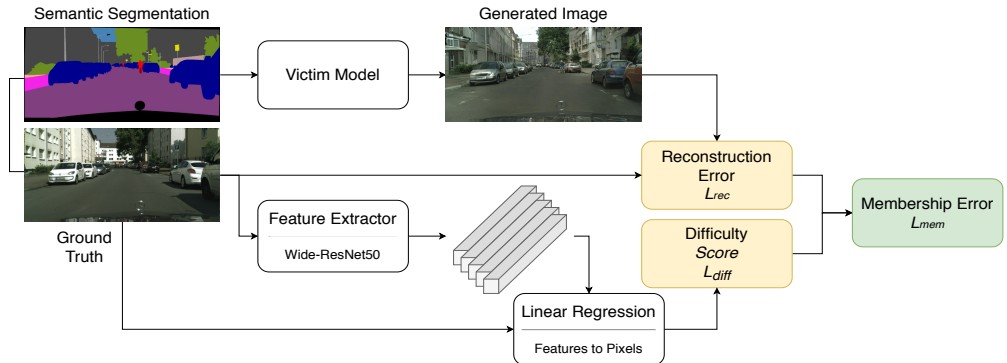

Figure 2: Illustration of the proposed black-box membership inference attack on conditional segmentation-to-image generation models. We would like to determine if a given image and its semantic segmentation were used in training. The victim model generates a reconstructed image based on the segmentation. In the top path the difference between the reconstructed image and the input image gives the reconstruction error $L_{rec}$. In the bottom path we compute the difficulty score $L_{diff}$ of the input image from the error of a linear predictor to predict pixel values of the ground-truth image from its deep features. Subtracting $L_{diff}$ from $L_{rec}$ gives the membership error.

## 4 EXPERIMENTS AND COMPARISONS

We conduct a thorough investigation demonstrating the effectiveness of our approach. First, we show its high success rate on various benchmarks. A comparison between our novel single-image, self-supervised, difficulty score to two alternative difficulty scores: a multi-image and a supervised difficulty scores, shows the superiority of our proposed method. We also compare our attack with the commonly used attack based on a shadow model and show that our attack is not only superior but also requires minimal assumptions over the attacker's knowledge. As MIA attacks are closely related to overfitting, we study the effect of overfitting on our attack success rate. Additional results as well as a discussion regarding possible defenses can be found in the appendix.

### 4.1 IMAGE TRANSLATION MEMBERSHIP INFERENCE ATTACK

We propose a novel membership attack on two popular image translation architectures - Pix2Pix (Isola et al., 2017) and Pix2PixHD (Wang et al., 2018), as well as three datasets - CMP Facades (Tyleček & Šára, 2013), Maps2sat (Isola et al., 2017) and Cityscapes (Cordts et al., 2016). All models are trained from scratch, with the exception of the Cityscapes dataset on the Pix2pixHD architecture in which we use the supplied large pre-trained model for computational constraints on the high resolution. In accordance with previous membership attack works, the success rate is measured using the ROC area under the curve (ROCAUC) metric. It can be seen in Table 1 that while using the reconstruction error alone achieves a high success rate, the membership error further improve these results by up to 6%. Fig 3 demonstrates the effect of subtracting the difficulty score from the prediction error. A single threshold on the membership error can separate train and test images. For more results, see Fig. 6 in the appendix.

We study the effect of utilizing common image augmentations, i.e. horizontal flipping and random cropping, over the pair $(x, y)$ in order to construct a larger set $\{(x_{aug}, y_{aug})\}$ and define $L_{rec}$ to be the average reconstruction error over the set. This can improve the accuracy by up to 5% on Pix2pix, see App. A.5 for details.

Possible defenses against our attack are discussed and evaluated in Sec. A.6 in the appendix.

### 4.2 SINGLE IMAGE DIFFICULTY SCORE

In this section we compare our single-image difficulty score (Sec. 3.2) with a multi-image difficulty score, in which a "shadow" model is trained on new data and define the difficulty score to be the

| | **Pix2pix** | | | **Pix2pixHD** | | |
|---|---|---|---|---|---|---|
| | Facades | Maps2sat | Cityscapes | Facades | Maps2sat | Cityscapes |
| Reconstruction error | 93.39% | 84.19% | 77.44% | 98.91% | 95.73% | 96.04% |
| Membership error | **96.62%** | **90.54%** | **82.23%** | **99.02%** | **99.89%** | **99.19%** |

Table 1: Membership attack ROCAUC using our (i) reconstruction error $L_{rec}$ and (ii) membership error $L_{mem}$. Using the membership error, which subtracts the image difficulty score from the reconstruction error, substantially improves performance.

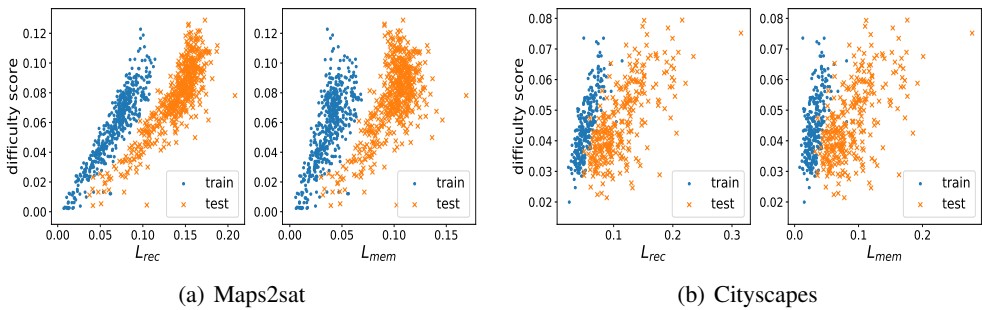

Figure 3: The proposed membership error $L_{mem}$ can better separate train and test images by a simple threshold (i.e. a vertical line) compared to the reconstruction error $L_{rec}$.

average $L_1$ reconstruction error on the new model. In order to upper-bound the multi-image difficulty score, we use the same shadow model architecture as that of the victim model (as this is the most favorable setting for it). We also ensure the shadow model's training data shares the same distribution as the victim's training data, by randomly sampling 100 images from the test set of the corresponding dataset. We did not perform this experiment on the Facades dataset as its test set does not contain a sufficient number of samples.

The results are presented in Table 2. In the Pix2pixHD model, the multi-image model badly overfit to its training data and is not able to generalize well enough to perform as a difficulty measure for other images. It is therefore inferior to our single-image difficulty score. In the simpler pix2pix model, the multi-image difficulty metric seems comparable to the single-image difficulty score, given a sufficient number of images. We hypothesize that it is due to the lower capacity of the architecture, which limits overfitting. This however comes at the cost of the often unrealistic requirement of extra training data. In addition, this assumes knowledge of the victim architecture. Our single-image score does not require extra training images or knowledge of the architecture, which makes it applicable in more cases. Fig 4 presents the effect of number of training images on the multi image calibration method. As can be seen, in the pix2pix model, at least 50 of the same distribution images are required in order to outperform our method. For Pix2pixHD none of the evaluated number of images outperformed our single-image score.

### 4.2.1 OUT-OF-DISTRIBUTION MULTI IMAGE

As suggested by He et al. (2019), we also compare our score to the more realistic scenario in which a large amount of similar but out-of-distribuition dataset is available. For this cause, we train a shadow model on $4K$ images from the BDD dataset (Yu et al., 2018), as done by He et al. (2019). We then use this model as a shadow model for a multi-image difficulty score to the Cityscapes dataset, as both datasets consist of street scene images and have compatible label spaces. Note that it assumes knowledge of the victim model, which is not always true and is therefore a best-case scenario. Table 2 demonstrates that this approach is inferior to our single-image difficulty score.

### 4.3 SUPERVISED DIFFICULTY SCORE

We compare our self-supervised single-image difficulty score with the supervised difficulty score described in Sec. 3.2. The supervised score was proposed by Tudor Ionescu et al. (2016), which defined image difficulty to be the human response time for solving a visual search task. In order to

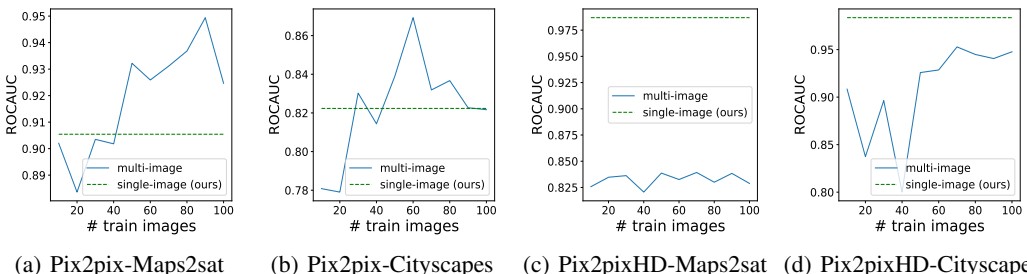

(a) Pix2pix-Maps2sat    (b) Pix2pix-Cityscapes    (c) Pix2pixHD-Maps2sat    (d) Pix2pixHD-Cityscape

Figure 4: Comparison of MIA accuracy when using our single image vs. using multi-image difficulty scores, as a function of the number of training images. Note that the multi-image score assumes knowledge of the victim's model, as well as the availability of many labeled training images.

| Model | Dataset | Single-Image | | Multi-Image | |
|-------|---------|------|------------|-------------|----------|
| | | Ours | Supervised | BDD Dataset | In-Dist. |
| Pix2pix | Facades | **96.62%** | 94.17% | - | - |
| Pix2pix | Maps2sat | 90.54% | 86.54% | - | **92.43%** |
| Pix2pix | Cityscapes | **82.23%** | 77.66% | 74.43% | **82.47**% |
| Pix2pixHD | Facades | **99.02%** | 98.86% | - | - |
| Pix2pixHD | Maps2sat | **99.89%** | 98.38% | - | 82.87% |
| Pix2pixHD | Cityscapes | **99.19**% | 96.86% | 66.2% | 94.76% |

Table 2: MIA accuracy of our method vs. using single and multi-image baselines for the difficulty score. Note that the BDD dataset is only relevant to Cityscapes and that in-distribution multi-image requires extra supervision of 100 images.

provide a fair comparison, we replace the pretrained VGG-f (Chatfield et al., 2014) features, used by Tudor Ionescu et al. (2016), with the more recent pretrained Wide-ResNet50 × 2 (Zagoruyko & Komodakis, 2016) features, as we use in our model. Fig. 13 in the appendix presents samples of images ranked as easy and hard by the supervised score. As can be seen in Table 2, our self-supervised single-image difficulty score outperforms the supervised difficulty score. Fig. 5 compares the relation between the reconstruction error and the supervised score to the relation between the reconstruction error and our self-supervised difficulty score, and shows that our score is better correlated to the reconstruction error. For comparison on other benchmarks, see Fig. 7 in the appendix.

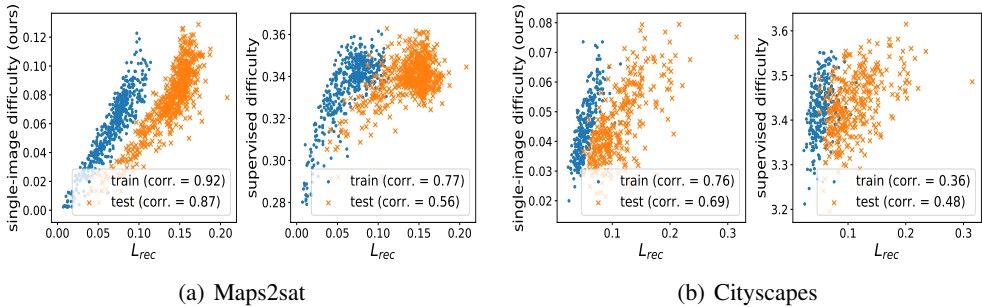

(a) Maps2sat          (b) Cityscapes

Figure 5: Comparison of the relation between the reconstruction error and the supervised difficulty score (right) to the relation between the reconstruction error and our self-supervised difficulty score (left) on Pix2pixHD. Our difficulty score is better correlated to the reconstruction error.

## 4.4 SHADOW MODELS

We compare our method with shadow-model-based methods, commonly used in membership inference attacks. A shadow model is trained on in-distribution data samples to create a labeled dataset

| Model | Dataset | Ours | Shadow Model - 100 | | Shadow Model - BDD | |
|-------|---------|------|------|------|------|------|
| | | | ROC | Acc. | ROC | Acc. |
| Pix2pix | Maps2sat | **90.54%** | 80.15% | 73.4% | - | - |
| Pix2pix | Cityscapes | **82.23%** | 78.68% | 67.5% | 72.57% | 56.16% |
| Pix2pixHD | Maps2sat | **99.89%** | 98.63% | 93.7% | - | - |
| Pix2pixHD | Cityscapes | **99.19%** | 96.39% | 64.0% | 95.78% | 56.5% |

Table 3: Comparison between our MIA and the commonly used shadow-model-based classifier attack, using 100 train and 100 test images, and the BDD shadow model. Our MIA outperforms while not requiring extra training images.

of known (train) and unknown (test) samples. The labeled dataset is then used to train a binary classifier to distinguish between the two. The assumption is that the classifier trained for membership attacks on the shadow model, will also detect membership of the target model.

We use the shadow models from Sec. 4.2, i.e. sharing the same architecture between the victim model $V$ and the shadow model $S$ and randomly sampling $N$ images from the test set, referred to as *shadow_train*, to train the shadow model $S$. A labeled dataset is constructed by randomly selecting additional $N$ test images, *shadow_test*. The training procedure is detailed in App. A.7.

We compare the ROCAUC of our attack against the classification accuracy and ROCAUC of the confidence score of the classifier $C$ when applied to images generated by the victim model $V$ in Table 3. Similar to He et al. (2019), we apply the classifier over several patches from the query image, and average the results over all patches. In order to provide a fair comparison we do not evaluate using the $2N$ images used in the training of the classifier. It can be seen that our attack outperforms the shadow model approach on all experiments. We further investigate the effect of the amount of training data used to train the shadow model and the corresponding classifier. As can be seen in Fig. 11, the success of the shadow model based attack depends on the amount of data used. This shows that in the common scenario of having just one or few images in-distribution images, the shadow model approach is not as effective as ours. As in Sec. 4.2.1, we compare to the more realistic scenario in which a large amount of similar but out-of-distribution dataset is available. We use the shadow model trained on $4K$ images from the BDD dataset (Yu et al., 2018) and train the classifier to distinguish between those $4K$ and additional $4K$ images that were not used in the training of the shadow model. As can be seen in Table 3, this approach is inferior to our single-image method.

## 4.5 EFFECT OF OVERFITTING

Membership inference attacks are closely related to overfitting in the victim model. In order to better understand this relation, we measure the success of our calibrated prediction error based attack under different levels of overfitting. We do so by evaluating our attack on checkpoints saved at different epochs during the training of victim model. For the purpose of this experiment, we also trained the Pix2pixHD model on the Cityscapes dataset instead of using the pretrained model. The results are presented in Fig. 9. We can observe that as the training process progresses, the victim model overfits more to the training data which results in higher attack success rates.

## 5 CONCLUSION

In this work, we present a black-box membership inference attack on conditional image generation models, e.g. image translation. At first we analyze a training-free attack based on the reconstruction error. We further improve this attack by proposing a novel image difficulty score, whose computation does not require an auxiliary training set. By utilizing this score, we can successfully discriminate between difficult images used in the training set, whose reconstruction error is large, and easy previously unseen images that have low reconstruction error. Our novel combined membership error was shown to achieve higher accuracy than baselines that use more supervision on multiple benchmarks.

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

## A  APPENDIX

### A.1  OTHER PRIVACY ATTACKS

Besides membership inference attacks, there exists a wide range of privacy attacks against neural networks. Model inversion attacks, first proposed by Fredrikson et al. (2014), aim at reconstructing features of the training data, e.g. recovering an image of a person from face recognition models (Fredrikson et al., 2015). Property inference attacks, proposed by Ganju et al. (2018), do not focus on the privacy of individual data samples, as in membership inference and model inversion attacks, but focus at inferring global properties of the training data, such as the environment in which the data was produced or the fraction of the data that comes from a certain class.

Model extraction attacks, also referred to as model stealing, attack a model $f$ by constructing a substitute model $\hat{f}$ that is either identical or equivalent to $f$ (Tramèr et al., 2016; Jagielski et al., 2020). Related line of work (Wang & Gong, 2018; Oh et al., 2019) attempts to infer hyperparameters such as the optimization proccess, e.g. SGD or ADAM.

### A.2  DETAILED DESCRIPTION OF OUR MIA ALGORITHM

Our MIA consist of computing the two terms in Eq. (3), i.e. $L_{rec}$ and $L_{diff}$ for a given query pair $(x, y)$, where $x$ is an image from the input domain and $y$ is the ground truth from the output domain, using only a black-box access to the victim conditional generation model $\mathbf{V}$.

$L_{rec}$ is computed using the $L_1$ error between the output image predicted by the model, $\mathbf{V}(x)$, and the ground truth image $y$, see step 1 in the Algorithm 1.

$L_{diff}$ is computed as the average error of a linear regression model, $\mathbf{P}$, in predicting pixel values from deep features of the ground truth image.

Our deep features are the activation values in the first 4 blocks of a pre-trained Wide-ResNet50$\times$2 (Zagoruyko & Komodakis, 2016). These features are of sizes $56\times56\times256$, $28\times28\times512$, $14\times14\times1024$, and $7\times7\times2048$. We interpolate all features to size $56\times56$ using bi-linear interpolation (step 2), and also reduce the input image to $56\times56$ (step 3). This gives a concatenated feature vector of size 3840 for each pixel $i$ in the $56\times56$ image ($256+512+1024+2048=3840$). We denote the concatenated feature vector for pixel $i$ as $\psi(i)$.

We randomly select $70\%$ of the pixels as train set, and compute a linear model $\mathbf{P}$ to estimate the RGB pixel values $y^i_{train}$ from the corresponding deep features $\psi_{train}(i)$ (step 4). The remaining $30\%$ of pixels will be used as a test split, $\{\psi_{test}, y_{test}\}$ (step 5). I.e. $|\psi_{train}| = 2195\times3840, |y_{train}| = 2195\times3$ and $|\psi_{test}| = 941\times3840, |y_{test}| = 941\times3$.

The linear regression model $\mathbf{P}$, a matrix of size $3840\times3$, is trained to minimize the error over $\{\psi_{train}, y_{train}\}$ (step 6). $L_{diff}$ will be the average absolute error over $\{\psi_{test}, y_{test}\}$ (step 7). We found that fitting the linear model to 70% of pixels and measuring the error on the remaining 30% gives better results than just measuring the error of the linear fitting.

We compute $L_{mem}$ according to Eq. (3) and compare the results with a predefined threshold value $\tau$, such that any pair $(x, y)$ for which is holds that $L_{mem}(x, y) < \tau$ is denoted as a member of the victim models' $\mathbf{V}$ train set (steps 8-9).

---

**Algorithm 1.** Membership Inference Attack
**Input:** Query pair $(x, y)$, victim model $\mathbf{V}$, feature extractor $\mathbf{F}$, scalar $\alpha$, threshold $\tau$
**Output:** Membership inference result

1. $L_{rec} = \|\mathbf{V}(x) - y\|_1$
2. $\psi = \mathbf{F}(y) // |\psi| = 56 \times 56 \times 3840$
3. $y = resize(y, 56 \times 56 \times 3)$
4. $\{\psi_{train}, y_{train}\} \xleftarrow{70\%} \{\psi, y\}$
5. $\{\psi_{test}, y_{test}\} = \{\psi, y\} \setminus \{\psi_{train}, y_{train}\}$
6. Train linear regression $\mathbf{P}$ with $\{\psi_{train}, y_{train}\}$
7. $L_{diff} = \frac{1}{N} \sum_{i=1}^{N} \|\mathbf{P}\psi_{test}(i) - y^i_{test}\|_1 // N = 941$
8. $L_{mem} = L_{rec} - \alpha \cdot L_{diff}$
9. **if** $L_{mem} < \tau$ **then**
       Return **True**
   **else**
       Return **False**

---

### A.3 PARAMETER SELECTION

We experimented with different values for the $\alpha$ value in Eq. (3). As can be seen in Fig. 8, $\alpha = 0.5$ was the best choice over all benchmarks.

### A.4 OVERFITTING

As mentioned in Sec. 4.5, overfitting is the main reason for the success of our method. Fig. 9 shows the accuracy of our method as a function of the number of epochs used for training the victim model, clearly suggesting that overfitting is indeed the vulnerability.

### A.5 EFFECT OF AUGMENTATION

In order to further increase the accuracy of our reconstruction-error-based attack, we take advantage of the augmentations that were used as part of the training process of the victim model. We use two common augmentations: random flip and random crop. We randomly sample $T$ augmentations, and estimate the prediction error as the average $L_1$-reconstruction error over all the augmentations. We note that for Pix2pixHD (Wang et al., 2018), we only used the flip augmentation, as random cropping is not used during its training process. More generally, using augmentations at inference time that not used for training produces corrupted results. Therefore in case of Pix2pixHD, $T = 1$ as we only add augmentation, the horizontally flipped query image. In case of Pix2pix, $T = 100$ was used. As can be seen in Table 4, utilizing augmentations on pix2pix enables us to improve the attack success rate by up to $5\%$.

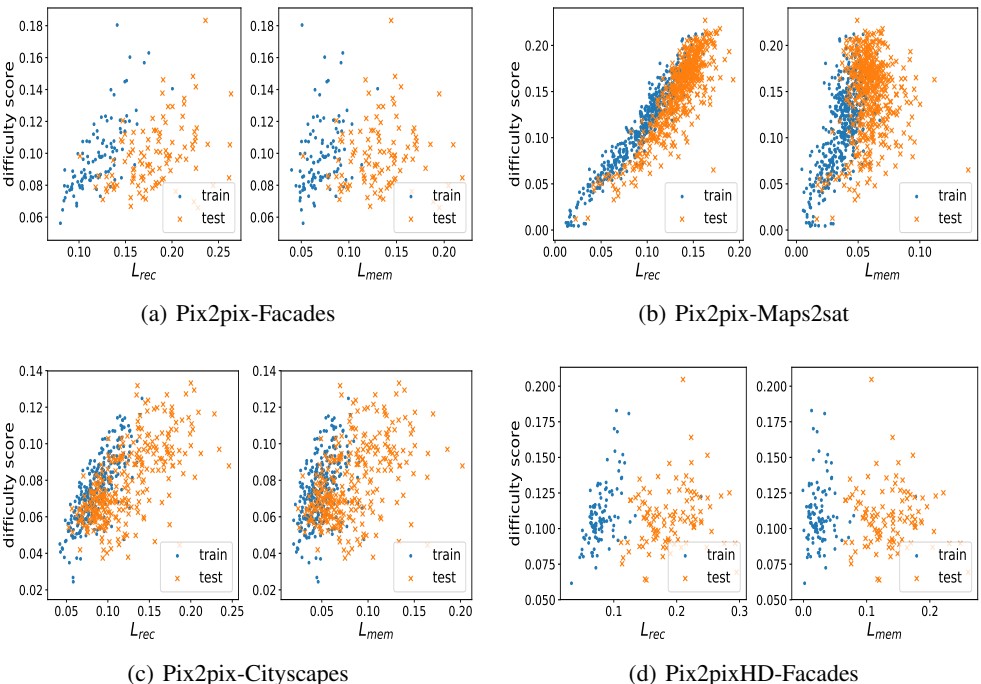

(a) Pix2pix-Facades  (b) Pix2pix-Maps2sat

(c) Pix2pix-Cityscapes  (d) Pix2pixHD-Facades

Figure 6: Using the membership score, subtracting the difficulty score from the reconstruction error, makes train and test sets better separated by a vertical line on the $x$ axis. Pix2pixHD for Maps2sat and Cityscapes are presented in Fig. 3

|  | **Pix2pix** | | | **Pix2pixHD** | | |
|---|---|---|---|---|---|---|
|  | Facades | Maps2sat | Cityscapes | Facades | Maps2sat | Cityscapes |
| Ours-base | 96.62% | 90.54% | 82.23% | 99.02% | 99.89% | 99.19% |
| Ours-Aug | **98.46%** | **92.21%** | **87.81%** | **99.09%** | **99.21%** | **99.91%** |

Table 4: Augmentations improve the accuracy of our attack.

## A.6 DEFENSES

Several defenses against membership inference attacks have been previously studied. As most previous works on MIA targeted classification models, not all defenses can be applied to conditional generation models. One such defense is the argmax defense, in which the victim model returns only the predicted label, rather then the full probability vector.

Other defenses attempt to reduce overfitting, which is highly correlated with MIA success rates. This can be done for example by changing the dropout or weight normalization ratios during the training of the victim model. Due to the computational resources required for training multiple models with different ratios, we do not evaluate our attack success against this defense, and refer the reader to Sec. A.4 for a visualization of the effect of overfitting on our attack, by attacking the victim model at earlier stages of the training process.

Two additional possible defense mechanisms are the differential private SGD (DP-SGD) and Gauss defenses. In DP-SGD (Abadi et al., 2016), the commonly used stochastic gradient descent optimization algorithm is modified in order to provide a differentially private (Dwork et al., 2014) model. This is done by adding Gaussian noise to clipped gradients for each sample in every training batch. However, this comes at the cost of computational complexity and model generation quality. There exists a trade-off between privacy and utility, in which the amount of added noise must be large enough to ensure privacy while not degrading the model's outputs to the point where the model is useless. Training a deep model, such as conditional generation models, with DP-SGD is an unsta-

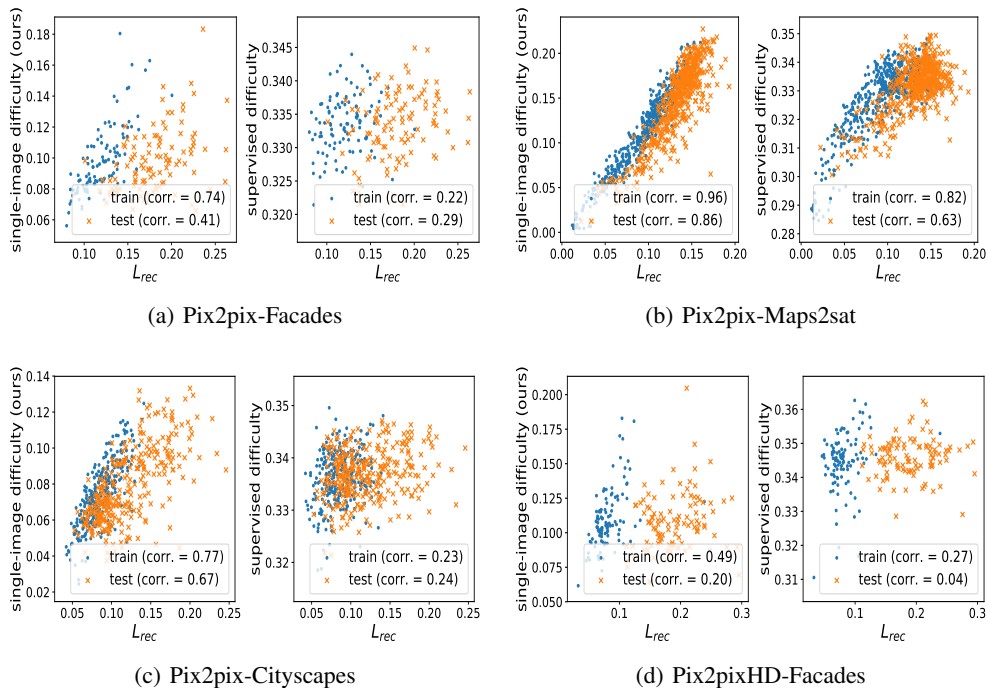

(a) Pix2pix-Facades

(b) Pix2pix-Maps2sat

(c) Pix2pix-Cityscapes

(d) Pix2pixHD-Facades

Figure 7: Comparison of the relation between the reconstruction error and the supervised difficulty score (right) to the relation between the reconstruction error and our self-supervised difficulty score (left). Maps2sat and Cityscapes are presented in Fig. 5. As can be seen, our score is better correlated to the reconstruction error.

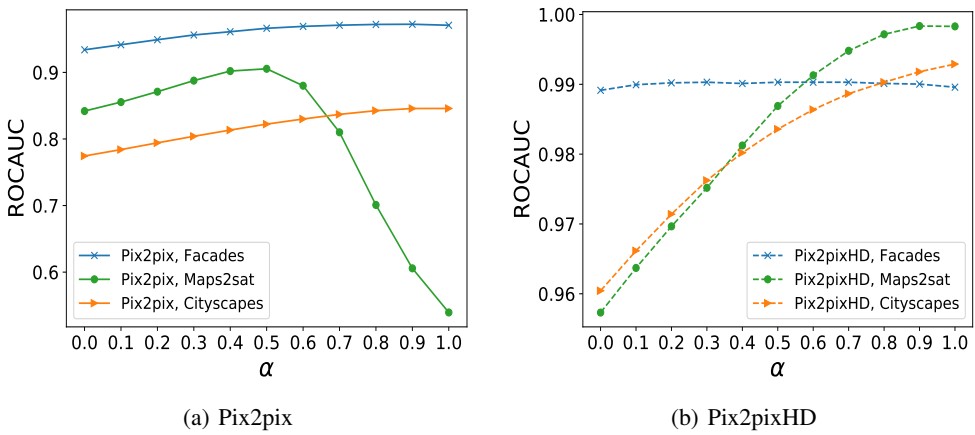

(a) Pix2pix

(b) Pix2pixHD

Figure 8: Effect of $\alpha$ in Eq. (3) over the attack success.

ble process. We experimented with multiple common configurations, i.e. added noise ratios and maximal gradient clipping threshold, and were not able to find a configuration that yields visually satisfying results. Hence, although the DP-SGD defense is theoretically protecting the victim model against membership inference attacks, in practice we find it to be impractical against our attack as it results with total corruption of the victim model.

In the Gauss defense, we add Gaussian noise to the image generated by the victim model (Gilmer et al., 2019). This attempts to hide specific artifacts of the overfitted model. We evaluate our attack success as a function of different noise STD. Fig. 10 shows that a considerable amount of noise,

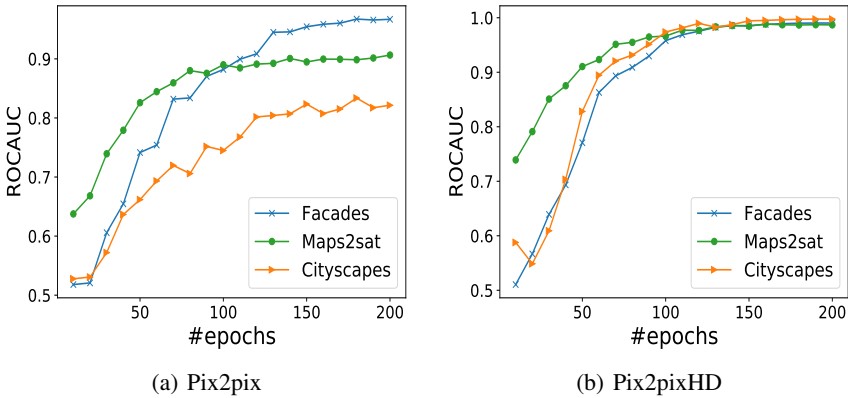

Figure 9: Effect of overfitting on the attack success rate. (a) Pix2pix, (b) Pix2pixHD

which corrupts the generated image, is required in order to have a significant effect over our attack success. Moreover, it can be seen that even with large amounts of noise, our attack still manages to succeed much better than random guessing. This implies that our attack is robust to the Gauss defense.

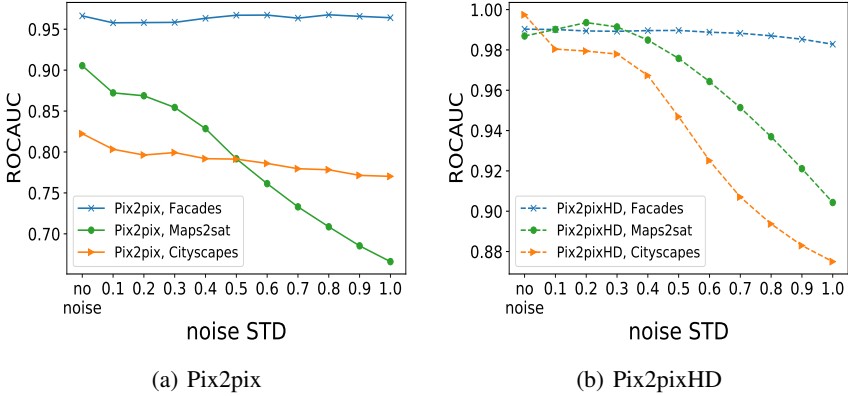

Figure 10: Effect of Gauss defense on the attack success rate. (a) pix2pix, (b) Pix2pixHD

### A.7 SHADOW MODEL TRAINING

After having selected $N$ images, denoted as *shadow_train*, which we used to train the shadow model and another $N$ images, *shadow_test*, which are not seen by the shadow model, we set the labels as:

$$label(x) = \begin{cases} 0, & \text{if } x \leftarrow shadow\_train \\ 1, & \text{if } x \leftarrow shadow\_test \end{cases} \tag{4}$$

The classifier **C** architecture and training procedure are similar to He et al. (2019). For each image, we compute the structured loss map between the ground-truth image and the generated image, and at every epoch we randomly crop 15 patches of size $90 \times 90$ from the structured loss map. We train a ResNet-50 (He et al., 2016) from scratch on the $90 \times 90$ patches, modified for binary classification. We use a batch size of 8, SGD optimizer, weight decay of $1e - 2$, initial learning rate of $0.1$ which reduces by a factor of $0.1$ every 15 epochs. As previously mentioned, we do not evaluate this on the Facades dataset, due to its size.

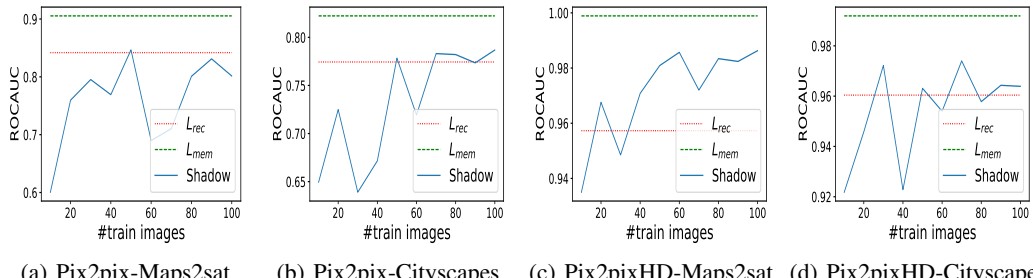

(a) Pix2pix-Maps2sat  (b) Pix2pix-Cityscapes  (c) Pix2pixHD-Maps2sat  (d) Pix2pixHD-Cityscape

Figure 11: Effect of the number of training images on shadow-model-based attacks. The shadow-model-based attack depends on the number of images used, we outperform this approach with no additional data.

## A.8 ADDITIONAL DATASETS

In In addition to the Facades, Maps2sat and Cityscapes datasets, we evaluated our attack over the Edges2shoes dataset (Yu & Grauman, 2014) and the CelebA dataset (Liu et al., 2015).

### A.8.1 EDGES2SHOES

The Edges2shoes dataset (Yu & Grauman, 2014) consists of 50K train images and 200 test images. Due to the large training set, the Pix2pix model has shown less overfitting to the data, and generates results that are visually inferior to the other datasets. The lack of overfitting results with our attack not managing to do better then a random guess, due to the strong correlation between overfitting and membership inference attacks. As Pix2pixHD has larger capacity, it managed to produce better results, in terms of visual quality, over this dataset. This also comes at the cost of more overfitting, as can be observed by the success of our attack over this model: $87.52\%$. This is an improvement of $5.5\%$ from our base attack, using only the reconstruction error, which achieves $81.99\%$.

### A.8.2 CELEBA

The CelebA dataset (Liu et al., 2015) is large-scale face attributes dataset with more than 200K celebrity images. We evaluated our attack on the CelebA dataset - by randomly sampling 10K images, extracting face landmark (Bulat & Tzimiropoulos, 2017) and training a pix2pixHD model to generate real face images from facial landmarks, see Fig. 12. Our base attack, using reconstruction error alone, achieves a success rate of $98.83\%$. By using our difficulty score, our attack further improves to a success rate of $99.04\%$. Faces are considered to be sensitive information, and the ability to accurately determine if a specific facial image was used to train a conditional generative model is a serious and concerning privacy breach.

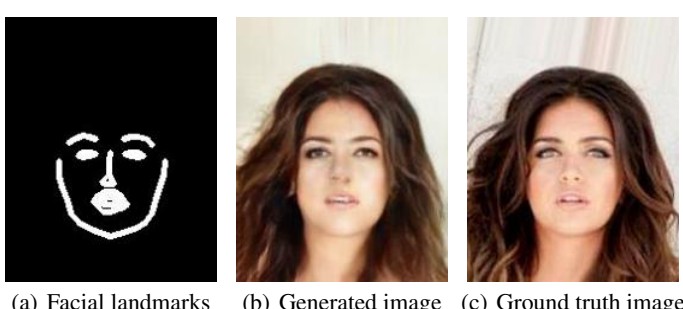

(a) Facial landmarks  (b) Generated image  (c) Ground truth image

Figure 12: Evaluating our attack on the CelebA dataset, by training a pix2pixHD model from facial landmarks to face images, achieves a high success rate of $99.04\%$.

## A.9    Supervised Image Difficulty Score

In Fig. 13, we present images ranked from hard to difficult using our implementation of the supervised-image difficulty score by Tudor Ionescu et al. (2016), for the Cityscapes and Maps datasets. The ranking seems correlated with image sharpness and level-of-detail images.

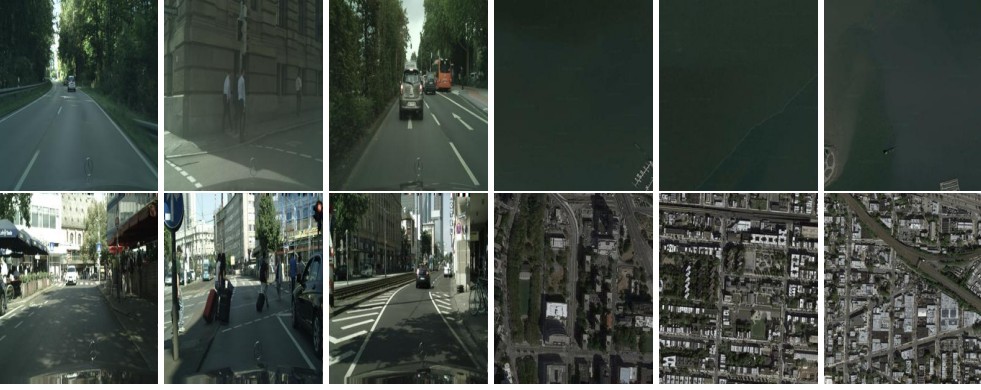

Figure 13: Examples of images from the Cityscapes (first two rows) and Maps2sat (last two rows) datasets that received the lowest (first and third row) and highest (second and last row) difficulty scores using the supervised difficulty score.

## A.10    ImageNet Difficulty Score

Our difficulty score relies on learning a mapping between feature vectors to their corresponding pixel values. We use a pre-trained Wide-ResNet50×2 (Zagoruyko & Komodakis, 2016), which is trained on the ImageNet dataset. We do not make any assumptions regarding an overlap between the pre-trained model's training data (i.e. ImageNet) and the data during in the attack. In the scenario in which such an overlap exists, the concern is that the difficulty score would lose its credibility.

In order to verify this, we computed the difficulty score of a random subset of 1K train images and 1K test images, from the ImageNet dataset. We do not observe any significant difference between the two - both share similar mean and std values: (0.0549, 0.018) for the train images and (0.0556, 0.0191) for the test images. A ROCAUC score of 51% further demonstrates that there is no clear difference between the distribution of the difficulty score on seen and unseen images.

Fig. 14 further demonstrates this. The first row presents the train images that received the lowest scores, i.e. marked as easy images, and the second row presents the test images with the lowest scores. Both correspond to "plain" images, regardless of whether they are known (train) or unknown (test). The same applies to the Difficult images. The third row presents the highest scored train images and the last row presents the highest scored test images. Both contains complex patterns and high variance. This demonstrates that the difficulty score is not affected by the having prior knowledge of the image, and is only measuring the amount of variance and complexity of an image.

## A.11    Conditional Difficulty Score

We evaluated the effect of conditioning the difficulty score computation on the data the victim model is conditioned on, i.e. training the regression model to predict pixel values from feature vectors extracted from the input to the conditional generative model. Both methods achieve very similar results on the Cityscapes dataset and pix2pixHD model: 99.02% when training on ground-truth vs. 98.82% when training on the segmentation mask. As the results are similar, with the ground-truth method being slightly superior, we chose to not be dependent of the input image and only use the ground truth image to estimate the image's difficulty.

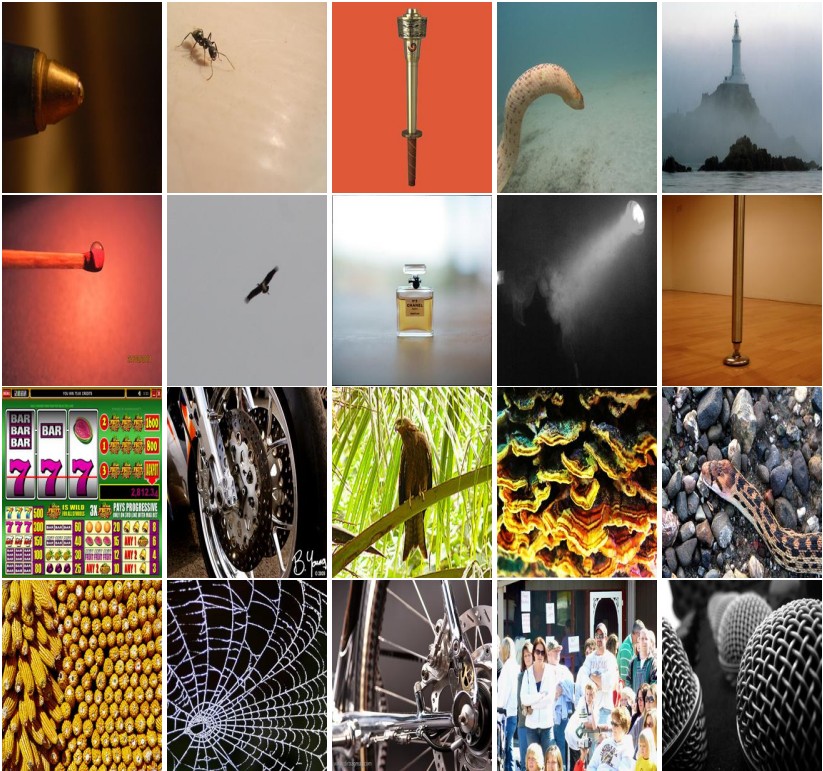

Figure 14: Examples of images from the ImageNet dataset that received the lowest and highest difficulty scores. First row - lowest scored train images. Second row - lowest scored test images. Third row - highest scored train images. Last row - highest scored test images. As can be seen, the difficulty score is effective even on images that were used for training the feature extractor.

## A.12 EFFECT OF DATA PARTITION IN DIFFICULTY SCORE

We evaluated the effect of the different partitions in the training of our difficulty score's regression model, on the Cityscapes dataset and pix2pixHD model. As can be seen in Fig. 15, training on less than $50\%$ of the image pixels results with unstable performance, while using $50\%$ or above is sufficient to achieve high success rates, with no preference to a specific value.

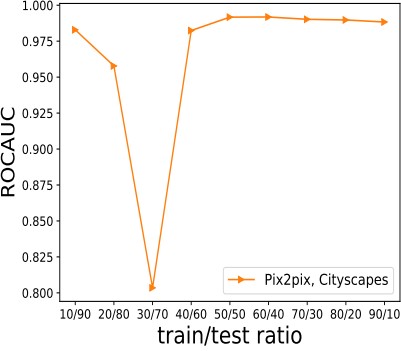

Figure 15: Effect of different data partitions in the training of the difficulty score's regression model, evaluated on the Cityscapes dataset and pix2pixHD model. Using less then $50\%$ of the image pixels results with unstable performance, while all values of $50\%$ or above result in similar attack success rates.

## A.13 EFFECT OF RESIZE METHOD

In the computation of the difficulty score, the ground truth image is resized using bicubic interpolation to an image of size $56 \times 56$ in order to match the $56 \times 56$ feature vectors extracted by concatenating the first $4$ blocks of a Wide-ResNet50×2 (Zagoruyko & Komodakis, 2016). We compared the effect of using different resize methods on the Cityscapes dataset for the pix2pixHD model. As can be seen in Fig. 16, the attack success rate is not very sensitive to the resize method.

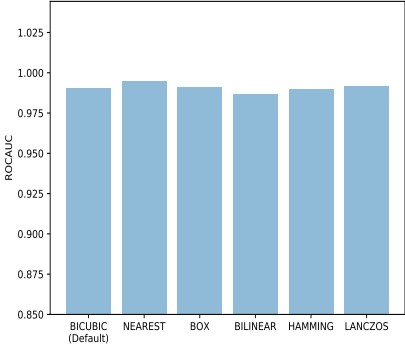

Figure 16: We evaluated the effect of different resize methods in the difficulty score on the Cityscapes dataset and pix2pixHD model. The attack success rate is not very sensitive to the resize method.

## A.14 EFFECT OF DIFFICULTY SCORE ON CONDITIONAL GENERATIVE MODEL TRAINING

We conducted initial experiments to explore the effect of using the difficulty score during the training process of the conditional generative model. We experimented with training pix2pixHD on the Cityscapes dataset while incorporating our difficulty score in the generator loss term.

The generator loss term, $L_G$ is composed of three terms: $L_{GAN}$ - the GAN loss term, $L_{FM}$ - similarity between features of the generator, and $L_{VGG}$ - perceptual similarity. We experimented with multiplying each one of these loss term with the difficulty score and multiplying all three term with the difficulty score, i.e. $L_G$. We compared the effect on our attack success rate, as a function of the progress in the training process, which we use to approximate different levels of overfitting.

Multiplying the loss value by the difficulty score encourages the generator to pay more attention to difficult images. As can be seen in Fig. 17, when this is applied on the GAN related loss terms, i.e. $L_{GAN}$ and $L_{FM}$, the model's overfitting slows down (we observe this by the slow down in the success of our attack, as it is highly correlated with overfitting). This happens as we pay more attention to the generator success against the discriminator and not to the reconstruction, which is mainly affected by $L_{VGG}$.

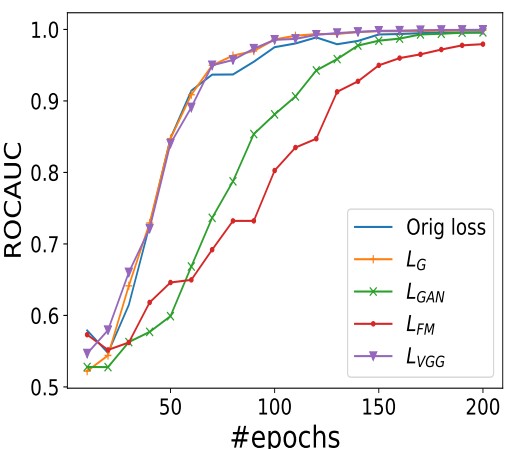

Figure 17: We evaluated the effect of incorporating our difficulty score in the training process of the conditional generative model. This encourages the model to pay more attention to difficult images. When we incorporate the difficulty score with the GAN related loss terms, the model's overfitting slows down, as demonstrated by the decrease in our attack success rate.

