# OpenReview forum: "Membership Attacks on Conditional Generative Models Using Image Difficulty"
_ICLR.cc/2021/Conference — Reject_

### Official Review · AnonReviewer1 · 2020-10-26
**Current version lacks a clear motivation and does not seem like a complete paper**

**Rating:** 5
**Confidence:** 3

**Review:**

The paper studies membership inference attacks (MIA) in the context of conditional image generation models, i.e., models learned for image-to-image translation. The problem aims to detect if a data sample is used to train a neural network model. The paper observes that some images are "universally" easy/difficult to reconstruct, and states using reconstruction error alone is less effective at discriminating between difficult images used in training and easy images that are never seen before. The paper designs a difficulty score to augment reconstruction error based method, and shows MIA results on multiple datasets.


The problem of MIA is meaningful in terms of detecting leakage of privacy-sensitive data, as stated in the introduction. But the motivation of this paper is not clear. The paper lacks an explanation why studying MIA in the context of image-to-image translation.

In the second paragraph of introduction, it is not clear either why MIA can "provide insights on the degree of overfitting in the victim mode"?

Section 3.2: What is the rationale behind the designed function Eq. (2) for measuring the difficulty of an image? The paper resizes all images into 56x56. Does this introduce noises that influences difficulty measurement? What is the resizing methods? Does different resizing methods have different performance?

Eq.(3): The paper considers "difficult" training images and "easy" unseen images, but does not consider "difficult" unseen images. For the unseen images that are difficult to reconstruction, does Eq.(3) still work?

Table 3 does not seem complete. Why are there not ROC and ACC for "ours"? What do the numbers mean under "ours"?

As the paper mentioned in related work that some prior work also studies MIA in the context of generative models (GANs and VAEs), it is natural to think what the unique conclusions through the study presented in this paper?

Figure 5: How to tell if "difficult score" is better correlated to the reconstrunction error than "supervised difficulty"?

--------
post-rebuttal
--------

I appreciate that authors have provided rebuttal that addresses many of my questions. I've read the updated paper and other reviewers' comments. The rebuttal has largely addressed my questions. However I am still not convinced by the following. Without a clarification I cannot see the significance of this work. So I'd maintain my rating.

I still don't get the motivation why studying MIA in image-to-image translation. I really wanted to solicit a motivational example. The authors merely say "image-to-image translation has yet to be studied in the context of MIA." I'm not convinced by this.

A.4 The authors simply draw curves of MIA performance vs. learning epoch. I still don't know where overfitting happens. From the context, it seems that the proposed method will fail if the victim model is not trained to be overfitting. In practice, surely victim models are not overfitting?

---

> ### Author Response · Authors · 2020-11-20
> **Response to Reviewer #1**
>
> We thank the reviewer for the dedicated review. The reviewer raised several valid points, which we believe can be satisfactorily addressed.
>
> **"The paper lacks an explanation why studying MIA in the context of image-to-image translation."**
> MIA attacks are studied in multiple contexts, e.g. classification models, generative models etc. The motivation is to study the potential data leakage in each such model, regardless of the specific dataset. Image-to-image translation has yet to be studied in the context of MIA. We also wanted to research the difference between this scenario to unconditional generative models, in which there is no input-output pair to compute the reconstruction error on. In such cases, the known black-box attacks are not able to achieve high success rates as our attack.
>
> **"In the second paragraph of introduction, it is not clear either why MIA can "provide insights on the degree of overfitting in the victim mode"?"**
> Figure 9, in Appendix A.4, demonstrates the effect of overfitting on the attack success rate. The more overfitted a model is, the more it is vulnerable to MIA.
>
> **"Section 3.2: What is the rationale behind the designed function Eq. (2) for measuring the difficulty of an image?"**
> Our method estimates the low level information extractable from the semantic features of the image. High prediction error means that the image is not easily specified using its high level attributes. This is an upper bound of the ability to specify the image from other semantic representations, such as segmentation masks, edges or landmarks. Therefore, the regression model's prediction error correlates with the difficulty of performing conditional generation task on this image.
>
> **"The paper resizes all images into 56x56. Does this introduce noises that influences difficulty measurement? What is the resizing methods? Does different resizing methods have different performance?"**
> The images are resized using bicubic interpolation on all pixels that contribute to one pixel in the resized image, using the Pillow library. This was selected because the feature extractor, Wide-ResNet50x2, also resizes the images using Pillow.
> Following your question, we evaluated the effect of different resize methods in Figure 16. in Appendix A.13. The attack success rate is not very sensitive to the resize method.
>
> **"Eq.(3): The paper considers "difficult" training images and "easy" unseen images, but does not consider "difficult" unseen images. For the unseen images that are difficult to reconstruction, does Eq.(3) still work?"**
> Yes, Eq.(3) still holds. The separation is difficult only between difficult seen and easy unseen images.
> Difficult unseen images result in high reconstruction errors, as can be seen in Figure 3, and therefore will be easy to distinguish from train images, which have significantly lower reconstruction error. The large gap from all training images remains even after the addition of the difficulty score.
>
> **"Table 3 does not seem complete. Why are there not ROC and ACC for "ours"? What do the numbers mean under "ours"?"**
> Our attack is based on the membership error score, and is evaluated using the  ROCAUC metric. As we do not train a classifier, we don’t measure accuracy. Therefore, the numbers under “ours” correspond to the AUCROC score. We compare our attack to the shadow model approach, which trains a classifier, so in this case we present both the classifier’s accuracy and the AUCROC score of the confidence values of the classifier (but only ROCAUC is directly related to MIA).
>
> **"As the paper mentioned in related work that some prior work also studies MIA in the context of generative models (GANs and VAEs), it is natural to think what the unique conclusions through the study presented in this paper?"**
> The difference between unconditional (others) and conditional (ours) generative models is the existence of input-output pairs in the black box setting. The existence of such pairs allows us to measure the reconstruction error. Unconditional generators do not have such pairs and cannot measure reconstruction. The same applies to VAE as only the generator is retained at inference time. As unconditional models do not have access to this knowledge, they attempt to approximate the reconstruction error by suggesting different alternatives to  “input-output” reconstruction, e.g. measuring the distance from target image to the nearest generated image or the number of generated images whose distance to target image is within some radius. These alternative methods are not able to obtain the high success rate that our attack achieves.
>
> **"Figure 5: How to tell if "difficult score" is better correlated to the reconstruction error than "supervised difficulty"?"**
> We measure the Pearson correlation between the reconstruction error and the difficulty score and present this in the “legend” of the plots presented in Figure 5.

---

### Official Review · AnonReviewer2 · 2020-10-27
**A simple and effective difficulty scoring function for MIA, experimental comparisons are comprehensive**

**Rating:** 6
**Confidence:** 3

**Review:**

This paper explores the MIA in the scenario of I2I translation and proposed a difficulty scoring function by observing that simply borrowing the MIA techniques from classification task is less effective. The scoring function is to measures the success of a linear regression model trained on part of pixels to predict the unseen pixel values. It is mainly to distinguish difficult-trained vs. easy-untrained examples. The improvement on performance brought by this simple function is obvious and comparisons are comprehensive. Below are some of my concerns and suggestions:

(1) The data used for training the regression model is the same with the one used for training the I2I model? Does it mean we still need to access the original data after I2I model pre-training?

(2) Is the 70/30 partition of pixels for training the regression model working the best? It will be better to analyze this partition ratio and see how the MIA accuracy will be affected.

(3) Generally if the pre-trained model is more generalizable (e.g., trained with more data), I feel the MIA will be less useful. An interesting direction is to explore whether it will be more useful in few-shot case, not only to detect whether overfitting happens, but also to explore how to use MIA to avoid overfitting and make the model trained with less data more robust.

========

Update:

Thanks for the author's feedback that clarifies my concerns. Adding the difficulty score into training looks like a more promising future direction.

---

> ### Author Response · Authors · 2020-11-20
> **Response to Reviewer #2**
>
> We thank the reviewer for the dedicated review. The reviewer raised several valid points, which we believe can be satisfactorily addressed.
>
> **"The data used for training the regression model is the same with the one used for training the I2I model? Does it mean we still need to access the original data after I2I model pre-training?"**
> The regression model (difficulty score) is trained only on the single query image (although the feature extractor is pretrained on ImageNet), and does not need to access the original data.
>
> **"Is the 70/30 partition of pixels for training the regression model working the best? It will be better to analyze this partition ratio and see how the MIA accuracy will be affected."**
> Figure 15 in Appendix A.12 demonstrates the effect of the different partitions in the regression model training. Training on less than 50% of the image pixels results in unstable performance, while using 50% or above is sufficient to achieve high success rates, with no preference to a specific value.
>
> **"Generally if the pre-trained model is more generalizable (e.g., trained with more data), I feel the MIA will be less useful. An interesting direction is to explore whether it will be more useful in few-shot case, not only to detect whether overfitting happens, but also to explore how to use MIA to avoid overfitting and make the model trained with less data more robust."**
> It is indeed interesting to explore the effect of using the difficulty score during the training process. We experimented with training pix2pixHD on the cityscapes dataset with using the difficulty score in the generator loss term. We found that incorporating the difficulty score to the GAN’s loss term has slowed down overfitting during training. We added a more detailed description in Appendix A.14.

---

### Official Review · AnonReviewer3 · 2020-11-01
**Borderline paper with concerns on novelty and applications**

**Rating:** 6
**Confidence:** 3

**Review:**

This paper aims to solve the problem of membership attack, ie detecting if data samples were used to train a neural network for conditional image generation. The paper first proposes a simple but effective approach by using the reconstruction error. To address the issue that reconstruction error alone is less effective at discriminating between difficult images used in training and easy images that were never seen before, a difficulty score is further proposed, which can be computed for each image, and its computation does not require a training set. The eventual membership error, obtained by subtracting the difficulty score from the reconstruction error, is shown to be effective on several datasets.

Merits:
+ The paper is well written. I can easily get the main idea of the paper. And this work is well-motivated: some images are universally easy, and others are difficult.
+ The proposed method looks simple and largely reasonable and is consistent with the motivation.
+ Experiments have shown good results in terms of segmentation-to-image translation. The ablation study shows the effectiveness of the proposed difficulty score.

Issues:
- The novelty of the proposed algorithm is limited: the reconstruction error is quite standard, and the difficulty score is not well explained. I wonder why the difficulty score is designed as in (2). Please provide more thorough explanations and clarify how it differs from previous approaches.
- While the title of the paper is "conditional generative models", the experiments are only conducted on segmentation-to-image translation. It will be important to show how it works in other related applications.
- The membership error relies on a hyper-parameter $\alpha$ in (3). Why is this parameter universally good for different datasets? Could the author provide more insights into tuning $\alpha$ for different tasks?

--------------------------------------------------------------------
I have read the response, and my rating is not changed.

---

> ### Author Response · Authors · 2020-11-20
> **Response to Reviewer #3**
>
> We thank the reviewer for the dedicated review. The reviewer raised several valid points, which we believe can be satisfactorily addressed.
>
> **"The novelty of the proposed algorithm is limited: the reconstruction error is quite standard, and the difficulty score is not well explained. I wonder why the difficulty score is designed as in (2). Please provide more thorough explanations and clarify how it differs from previous approaches."**
> Our method estimates the low level information extractable from the semantic features of the image. High prediction error means that the image is not easily specified using its high level attributes. This is an upper bound of the ability to specify the image from other semantic representations, such as segmentation masks, edges or landmarks. Therefore, the regression model's prediction error correlates with the difficulty of performing conditional generation task on this image.
>
> **"While the title of the paper is "conditional generative models", the experiments are only conducted on segmentation-to-image translation. It will be important to show how it works in other related applications."**
> We evaluated our attack on two datasets in which the input is not a segmentation mask. Section A.8.1 in the appendix provides our results on the large edges2shoes dataset. As can be seen, our attack achieves high success rates - 87.52%. We also evaluated our attack on the celebA dataset - by extracting face landmarks and training the pix2pixHD model to generate the real face image from it’s landmarks. Due to computational constraints we trained on 10K images. As can be seen in section A.8.2 in the appendix, our attack achieves very high success rates on this dataset as well - 99.04%.
>
> **"The membership error relies on a hyper-parameter α in (3). Why is this parameter universally good for different datasets? Could the author provide more insights into tuning α for different tasks?"**
> As in section A.3, we found the value 0.5 to be the best choice on average over all benchmarks, without the need for specific tuning or more supervision. An optimal $\alpha$ can be computed for each victim dataset if we can have access to it. However, this is not the case in MIA.

---

### Official Review · AnonReviewer4 · 2020-11-02
**Interesting method**

**Rating:** 6
**Confidence:** 3

**Review:**

This submission focuses on the problem of membership inference attack for conditional image synthesis. This is an interesting new application and a meaningful extension of the topic of MIA. It is straightforward to use reconstruction loss as a measure for membership error. However, the authors pointed out that on top of reconstruction loss, the difficulty for the sample should also be taken into account. With that the authors propose a novel method of measuring generation difficulty for each sample by training a regressor to estimate pixel value using the abstract representation by a pre-trained model. The regressor is trained using a portion of randomly selected pixels and tested on the rest. The error of the prediction is use as the indicator of difficulty. This is an intuitive method and from the results of the experiment, the proposed method seems to be quite effective. The reviewer would like the author to clarify a few points: 1) is it an assumption that the pre-trained network has never seen any data used in the testing? If not, what would happen if the data has been seen when training the “feature extractor” but not the victim model? 2) Wouldn’t it be more application/problem-specific to train a regressor using the data the victim model is conditioned on (e.g. the segmentation mask) to predict the pixel values of the ground-truth image in estimating sample difficulty?

---

> ### Author Response · Authors · 2020-11-20
> **Response to Reviewer #4**
>
> We thank the reviewer for the dedicated review. The reviewer raised several valid points, which we believe can be satisfactorily addressed.
>
> **"is it an assumption that the pre-trained network has never seen any data used in the testing? If not, what would happen if the data has been seen when training the “feature extractor” but not the victim model?"**
> We do not make any assumption over the existence of an overlap between the ImageNet dataset, on which the pre-trained model is trained, and the dataset used in the attack or the victim model’s training.
> In order to verify the difficulty score’s validity on seen data, we investigated the difficulty scores of 1K train images and 1K test images from the ImageNet dataset. We do not observe any significant difference between the two - both share similar mean and std values: (0.055, 0.018) for the train images and (0.056, 0.019) for the test images. A ROCAUC score of 51% further demonstrates that there is no clear difference between the distribution of the difficulty score on seen and unseen images. Figure 14 in Appendix A.10 presents the images that received the highest and lowest scores. We did not observe a visible difference between the train and test images.
>
>
> **"Wouldn’t it be more application/problem-specific to train a regressor using the data the victim model is conditioned on (e.g. the segmentation mask) to predict the pixel values of the ground-truth image in estimating sample difficulty?"**
> We evaluated the effect of training the regression model on the segmentation mask rather than the ground truth image. Both methods achieve very similar results on the Cityscapes dataset and pix2pixHD model:  99.02% when training on ground-truth vs. 98.82% when training on the segmentation mask. As the results are similar, with the ground-truth method being slightly superior, we chose to use the more “general” method that only depends on the image and does not depend on the specific input.

---

### Decision · Program_Chairs · 2021-01-07
**Final Decision**

**Decision:**

Reject

**Comment:**

The work focuses on detecting whether a certain data sample was used to train a deep network-based conditional image synthesis model. The key idea is not to rely on just reconstruction error but normalizing it via a proposed difficulty score. The reviewers found the problem statement important and the paper easy to follow. However, a common concern in the discussion was the approach is specific to image-to-image translation problems. Many other minor questions were answered by the authors in the rebuttal. Upon discussion post rebuttal, the reviewers decided to maintain their score. AC and reviewers believe that the paper will benefit from better analysis and description of difficulty score and its correlation with reconstruction error. It would be ideal to see how these ideas are useful for a broader problem than image translation. Please refer to the reviews for final feedback and suggestions to strengthen the submission.